# Flow Behavior of Liquid Steel in Fewer Strands Casting of Six-Strand Bloom Tundish

**Xianyang Wang [1], Sijie Wang [1], Hao Hu [1], Xin Xie [2], Chenhui Wu [2], Dengfu Chen [1,\*] and Mujun Long [1,\*]**

1   College of Materials Science and Engineering, Chongqing University, Chongqing 400044, China
2   State Key Laboratory of Vanadium and Titanium Resources Comprehensive Utilization, Panzhihua 617000, China
*   Correspondence: chendfu@cqu.edu.cn (D.C.); longmujun@cqu.edu.cn (M.L.); Tel.: +86-13658333267 (D.C.); +86-13594318151 (M.L.)

**Abstract:** In continuous casting, it is common to close single or multiple submerged nozzles of multi-strands tundish to adapt to production rhythm due to insufficient liquid steel or equipment failure. However, the closure of the nozzle will change the flow field in the tundish and further affect the removal efficiency of inclusions in the tundish. For this reason, based on numerical simulation, the flow behavior of liquid steel and the removal of inclusion in tundish with different nozzle closed were studied, and the optimal nozzle closing scheme was obtained, which provided a basis for the selection of nozzle closing in tundish. At the same time, the gas curtain is set in the tundish to alleviate the negative effects such as the increase of dead zone caused by closing nozzle. The results show that the removal rate of inclusions with sizes 10, 30, 50, 70, and 90 μm change from 12.4%, 39.1%, 74.2%, 93.3%, and 95.6% to 14.7%, 36.4%, 76.4%, 85.3%, and 93.8%, respectively. The volume of the tundish dead zone is increased after closure of nozzle, the dead zone of the tundish is improved when the gas is installed, and the dead zone volume was reduced from 14.8% and 16.4% to 13.9 and 14.1%.

**Keywords:** multi-strand tundish; fewer strands casting; flow field; inclusion; residence time distribution; gas curtain





## 1. Introduction

With the rapid development of economy, modern high-speed railway, heavy haul railway transportation have seen unprecedented development. Therefore, the quality and performance of rail demand high requirements. High strength, high toughness, noble and pure purification are the main development directions with regard to railway. To satisfy the various requirements in the railway, developing high-quality steel is crucial; hence, steel composition and organization, and all kinds of the inclusions in steel and the key object of control need to be strictly controlled [1].

Tundish is a transition equipment that changes from intermittent production process to continuous production process in the process of continuous casting of steel, and its initial functions mainly include diverting effect, continuous pouring effect, decompression effect, and protection effect [1,2]. With the research of large number of metallurgical scholars and the development of metallurgical technology, tundish plays an efficient role in homogenizing the composition and temperature of liquid steel and removing non-metallic inclusions in liquid steel, which is of great significance for the high quality of steel.

With the continuous improvement in steel quality requirements, the removal of inclusions in tundish has become a hot research topic, and many metallurgical workers have carried out a lot of research on it [3–13]. Wang Y., et al. [14] discusses the inclusion transport and the influence of structural parameters of baffle holes on inclusion removal rate. It is concluded that reasonable tundish structure can significantly improve the removal of inclusions. Sinha AK, et al. [15] concluded that large particle inclusions (about 120 μm)

were more likely to enter the protective slag, while small particle inclusions (about 40 μm) usually could not float to the slag or steel interface. Sheng D., et al. [16] investigated four different tundish configurations and the effect of various parameters, such as the inclusion size, the inclusion density at the normal casting conditions, and the results show that the flow control devices reduce the extent of the dead volumes in the tundish and thus enhance the removal efficiency of the inclusions. Tkadlečková M, et al. [17] used the residence time distribution (RTD) curves and inclusion removal efficiency for evaluation of steady state steel flow character depending on the internal configuration of a tundish with an impact pad in two design modifications, and the results showed that the suitable tundish structure can improve the removal rate of inclusions significantly. Xie X., et al. [18] used hydraulic and mathematical simulation methods to study the effect of weir and air curtain on removing inclusions from the continuous casting tundish, and the results showed the optimum scheme got larger inclusion flotation rate.

Due to the strict requirement of quality stability of heavy rail steel, constant casting speed is required in field production. In on-site production, problems such as insufficient supply of liquid steel or equipment failure often occur. Therefore, in order to ensure the matching of continuous casting and refining cycles, one or more submerged nozzles need to be closed for temporary casting operation, which is usually called fewer strand casting [19]. However, the closure of the submerged nozzle will lead to changes in the tundish flow field and further affect the removal efficiency of the inclusions in the tundish. Therefore, it is very possible to have a great impact on the metallurgical function of the tundish by closing a certain submerged nozzle. Cheng Y., et al. [20] studied sixth-strand tundish in fewer strands casting, a temperature deviation index was introduced to characterize the temperature stratification of molten steel for a large capacity tundish, and a new calculation method of residence time curve was used to describe the different flow types of molten steel at each outlet, and the tundish structure is optimized. After investigation, few metallurgical scholars pay attention to the flow field and inclusion removal in the tundish with low flow casting, but it is undeniable that this may have an impact on the quality of the product.

Generally, the removal of inclusions in tundish can be judged indirectly by analyzing liquid steel residence time distribution curve (RTD) and liquid steel flow in water simulation and numerical simulation tests [19–22], and the inclusions and gas curtain (argon curtain) in tundish can also be directly simulated by DPM (discrete phase model) [18,23]. In this paper, the flow characteristics of tundish and the removal rate of inclusions are analyzed based on numerical simulation, and the optimal scheme of closing nozzle is obtained. The research work can provide a basis for manufacturers to adapt to the production rhythm of the choice of closing nozzle, in order to adapt to the situation of fewer strand casting.

## 2. Model Description

### 2.1. Geometric Models and Meshing

In this paper, 40 t bloom six-strands tundish is taken as the research object. The inner diameter of its ladle shroud is 90 mm, the insertion depth is 200 mm, and the inner diameter of the submerged nozzle is 40 mm. The specific size is shown in Figure 1. In the software ANYSY-Meshing, tetrahedral structure is used to divide the mesh. After the local mesh is encrypted, the generated tetrahedral mesh is 288,823, as shown in Figure 2.

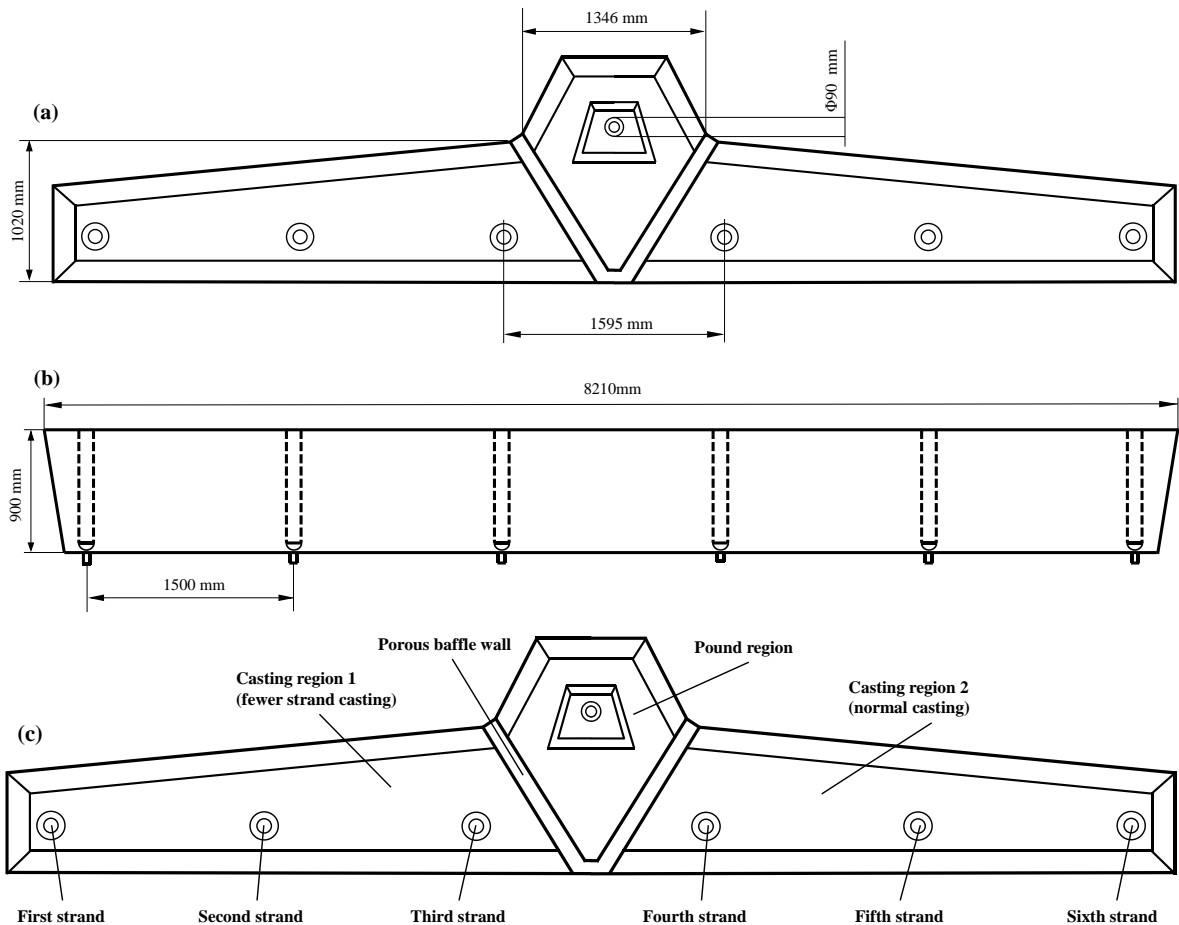

**Figure 1.** Diagram of tundish structure: vertical view (**a**); front view (**b**); schematic diagram of fewer strands casting (**c**).

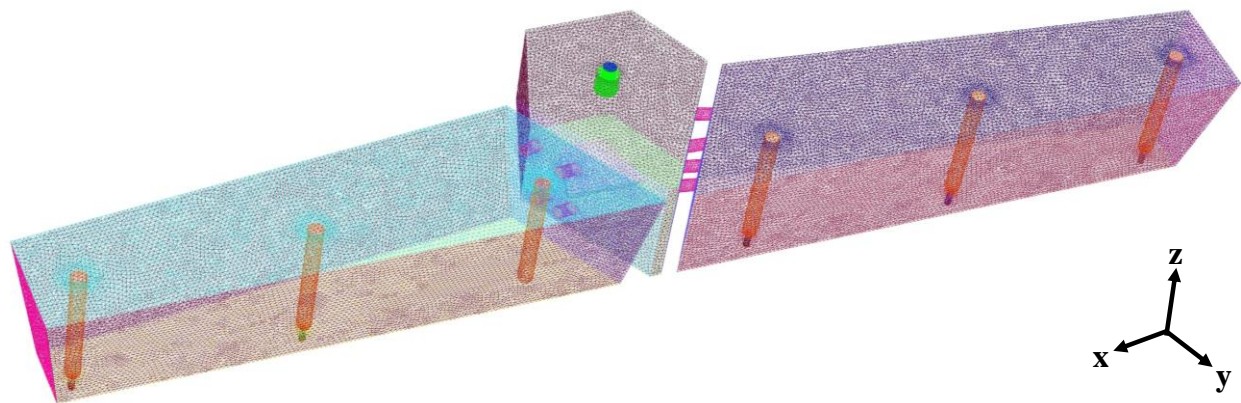

**Figure 2.** Mesh of numerical model of tundish.

### 2.2. Assumptions

In the actual industrial production, the flow of liquid steel in tundish is a very complicated turbulent flow, accompanied by complex reactions, there are too many unstable factors, the movement of inclusions is full of too much randomness, it is necessary to abstract the ideal model from the actual production. In order to facilitate the study of flow phenomena in continuous casting tundish, the following assumptions are made during the numerical simulation of tundish.

(1)    The influence of surface fluctuation and surface slag on flow of tundish is not considered.

(2)　The liquid steel flow is steady incompressible flow, and the fluid is driven by the initial velocity of the pure liquid phase.

(3)　Assume that the flow process in the tundish is steady state.

(4)　The temperature change in the whole process is not considered.

(5)　The motion of liquid steel belongs to turbulent flow with high Reynolds number.

(6)　The inclusion is assumed to be spherical in shape.

(7)　Regardless of the collision polymerization between the inclusion particles and the adsorption of the wall surface, it is determined that the inclusion in the z direction is absorbed by the slag layer when it reaches the liquid level.

(8)　The object of study is spherical nonmetallic inclusions in liquid steel.

(9)　The main way to remove inclusions in tundish is adsorption of liquid level slag.

### 2.3. Fundamental Equations

2.3.1. Fluid Flow

The mathematical model of liquid steel flow in tundish can be expressed by continuity equation and momentum equation.

Continuity equation is

$$\frac{\partial \rho}{\partial t} + \nabla \cdot \left( \rho \vec{v} \right) = S_m \tag{1}$$

where $\rho$ is fluid density, kg/m$^3$; t is flow time, s; $\vec{v}$ velocity vector, m/s; $S_m$ is source phase.

Momentum equation (Navier–Stokes equation) is

$$\frac{\partial}{\partial t} \left( \rho \vec{v} \right) + \nabla \cdot \left( \rho \vec{v} \vec{v} \right) = -\nabla p + \nabla \cdot \left( \bar{\bar{\tau}} \right) + \rho \vec{g} + \vec{F} \tag{2}$$

where $p$ is the static pressure; $\bar{\bar{\tau}}$ is the stress tensor; $\rho \vec{g}$, $\vec{F}$ are gravity and other forces on the fluid, respectively.

The $\bar{\bar{\tau}}$ expression is

$$\bar{\bar{\tau}} = \mu \left[ \left( \nabla \vec{v} + \nabla \vec{v}^T \right) - \frac{2}{3} \nabla \cdot \vec{v} \bar{\bar{I}} \right] \tag{3}$$

where $\mu_1$ is fluid viscosity, Pa·s; $\bar{\bar{I}}$ is unit tensor.

2.3.2. Motion of Inclusions

Discrete phase model is generally used for the motion of nonmetallic inclusions in liquid steel. In Lagrange coordinate system, Navier–Stokes equation and particle trajectory equation are solved simultaneously to determine the trajectory of nonmetallic inclusions in liquid steel.

The external forces on the inclusions moving in the liquid steel include: fluid drag force (viscous drag), gravity, buoyancy, pressure gradient force, false mass force, Basset force, Magnus force, Saffman lift, etc. For inclusions in liquid steel, fluid drag force on particles is the most important, followed by gravity, due to small particle size (micron magnitude) and thin concentration. Other forces are very small in magnitude and generally negligible.

The transient particle momentum equation established by the random trajectory model is shown in Equation (4) [18].

$$\frac{du_p}{dt} = F_D \left( \bar{u} - u_p \right) + \frac{g(\rho_P - \rho)}{\rho_p} + F_{other} \tag{4}$$

where $u_p$ is inclusion particle velocity, m/s; $F_D$ is fluid resistance of a particle per unit mass, detailed explanation can be found in the literature [18]; $\bar{u}$ is the velocity of liquid steel, m/s;

$g$ is the gravitational acceleration, m/s$^2$; $\rho p$ is the density of the inclusion particles, kg/m$^3$; $F_{other}$ is other interphase forces.

Fluid resistance $F_D$ is

$$F_D = \frac{18\mu}{\rho_p d_p^2} \frac{C_D Re}{24} \tag{5}$$

where $\mu_2$ is dynamic viscosity of liquid steel, kg/(m·s); $d_p$ is particle diameter, m; $C_D$ is drag coefficient; $R_e$ is the Reynolds number.

For smooth spherical particles, the drag coefficient is

$$C_D = a_1 + \frac{a_2}{Re} + \frac{a_3}{Re^2} \tag{6}$$

where $a_1$, $a_2$, and $a_3$ are constant, which apply to several ranges of $R_e$ given by Morsi, Alexander, can be seen in the literature for details [23].

### 2.4. Boundary Conditions

According to the sampling and testing [24], the inclusion of silicon killed steel in the pouring process of tundish is mainly silica, and the size range is mainly 10–90 μm. According to the actual flow of fluid in tundish, some boundary conditions are set up in numerical simulation.

(1) The condition of velocity-inlet is adopted at the inlet of tundish, and the velocity direction is downward. According to the casting speed and billet section, the inlet velocity of lad shroud is 1.1266 m/s. (2) The velocity-inlet condition is also adopted at the submerged nozzle of the tundish. A negative value indicates the opposite velocity direction. The total discharge at the outlet is equal to that at the inlet, and the velocity at each outlet is set to 0.9703 m/s. (3) The upper liquid surface was set as the free sliding wall surface, and the shear force was set as 0. (4) The bottom and side of the tundish are non-sliding wall surfaces. The standard wall function is used near the wall surface, and the normal gradient is 0.

For the DPM model simulating inclusions, the boundary conditions are as follows: (1) The inlet is set as the escape boundary, the particles are uniformly arranged, and enter the tundish at the same speed as the steel liquid phase; (2) the outlet is set as the escape boundary; (3) the free liquid level is set as the trap boundary; (4) the tundish wall, weir, dam, and other walls are set as the reflect boundary.

For argon in the gas curtain, DPM model was adopted, and the bubble size was 5 mm. Specific numerical simulation parameters are shown in Table 1.

**Table 1.** Numerical simulation parameters.

| Parameters | Value |
| --- | --- |
| Density of liquid steel/(kg·m$^{-3}$) | 7020 |
| Density of inclusion/(kg·m$^{-3}$) | 2200 |
| Viscosity of liquid steel/(kg·m$^{-1}$·s$^{-1}$) | 0.0062 |
| Inlet velocity/(m·s$^{-1}$) | 1.1266 |
| Gravitational acceleration/(m·s$^{-2}$) | 9.81 |
| Density of argon/(kg·m$^{-3}$) | 1.6228 |
| Viscosity of liquid steel/(kg·m$^{-1}$·s$^{-1}$) | 0.0000212 |

### 2.5. Calculation of Inclusion Removal

The removal rate of inclusions is calculated according to the simulation results of DPM model, and its expression is

$$T_a = \frac{N_{trap}}{N_{in}} \times 100 \tag{7}$$

where $N_{trap}$ is the number of inclusions captured; $N_{in}$ is the number of inclusions at the inlet.

### 2.6. Average Residence Time of Tundish

In this paper, the theoretical average residence time is calculated by the ratio of steel capacity to steel flux. On this basis, the modified classical combination model mentioned in the literature [25] is adopted to calculate the volume fraction of dead zone, plug zone, and well-mixed zone in the tundish.

The dead zone volume fraction is

$$\frac{Vd,i}{V} = \int_2^\infty Ei(\theta)d\theta \tag{8}$$

$$\frac{Vd}{V} = \sum_{i=1}^n \frac{Vd,i}{V} \tag{9}$$

The well-mixed zone volume fraction is

$$\frac{Vba,i}{V} = \sigma^2 i - \frac{Vd,i}{V} - \frac{Vby,i}{V} \tag{10}$$

$$\frac{Vba}{V} = \sum_{i=1}^n \frac{Vba,i}{V} \tag{11}$$

The plug zone volume fraction is

$$\frac{Vp,i}{V} = \frac{1}{n} - \frac{Vba,i}{V} - \frac{Vd,i}{V} - \frac{Vby,i}{V} \tag{12}$$

$$\frac{Vp}{V} = \sum_{i=1}^n \frac{Vp,i}{V} \tag{13}$$

where $\frac{V_{d,i}}{V}$ is the ratio of the $i$'th flow dead zone of the tundish; $Ei(\theta)$ is the residence time distribution function of the $i$'th flow; $\theta$ is time; $\frac{V_d}{V}$ is the ratio of the total dead zone of the tundish; $\frac{V_{ba,i}}{V}$ is well-mixed zone ratio of $i$'th flow; $\sigma^2$ is the variance of the residence time distribution; $\frac{V_{by,i}}{V}$ is the short-circuit flow ratio of i'th flow; $\frac{V_{ba}}{V}$ is the ratio of total well-mixed zone of tundish; n is the total strands number of the tundish; $\frac{V_{p,i}}{V}$ is the proportion of $i$'th flow plug zone in tundish, $\frac{V_p}{V}$ is the ratio of the total plug zone of tundish.

### 2.7. Simulation Scheme

In order to get the best closure scheme, considering the symmetry of tundish, several different schemes are set up, and the results are shown in Table 2. The closed submerged nozzle in the scheme is shown in Figure 1.

**Table 2.** Scheme of fewer strands casting.

| Original Scheme | Scheme 1 | Scheme 2 | Scheme 3 |
|---|---|---|---|
| Normal casting | Only close first strand | Only close second strand | Only close third strand |

The installation scheme of air curtain is shown in Table 3 and Figure 3.

**Table 3.** Installation scheme of air curtain.

| Parameter | Position 1 | Position 2 | Position 3 |
|---|---|---|---|
| Distance from the air curtain to the center of the ladle shroud, mm | Under porous baffle wall | 1200 | 1750 |
| argon flow, L/min | 12 L/min | 12 L/min | 12 L/min |

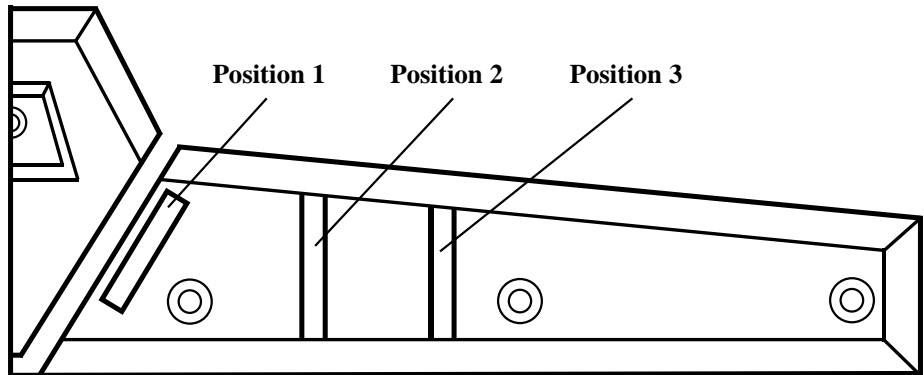

**Figure 3.** Position of gas curtain.

## 3. Results and Discussion

### 3.1. Analysis of Tundish RTD Curve and Flow Field in Normal Casting

During normal casting, the flow field of the tundish is symmetrical. At this time, the RTD curve and liquid steel flow line of the tundish are shown in Figure 4. As can be seen from Figure 4a, the first strand of the billet tundish has the same overlap degree with the third strand curve, while the second strand has a poor overlap degree with the other two flows. There is a big difference in peak time and peak concentration, and the width of the peak area is very narrow, which indicates that there is an obvious short circuit flow in the flow of liquid steel in the tundish. The response time of the second strand is very short, which is 16 s, much less than the 66 s of the third strand. The short response time is not conducive to the removal of inclusions. According to the RTD curve, the average residence time of the liquid steel at the three submerged nozzles is 806 s, 648 s, and 705 s, respectively, and the dead zone volume fraction of each flow is 10.9%, 8.5%, and 8.0%, respectively. According to the calculation method of the population RTD curve of the multi-strand tundish, the average residence time of the tundish is 716 s.

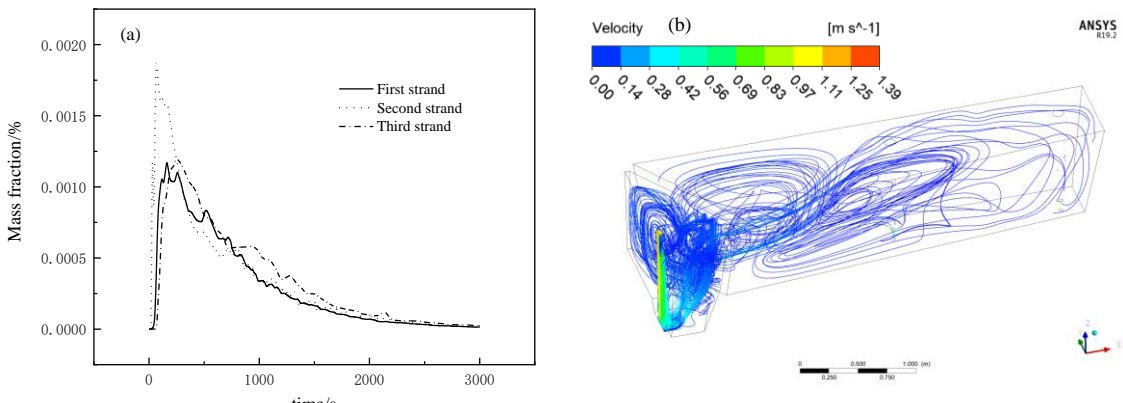

**Figure 4.** RTD curves of tundish for normal casting and liquid steel flow diagram: RTD curve (**a**); liquid steel flow diagram (**b**).

As can be seen from Figure 4b, in the normal pouring process of tundish, the liquid steel entering through the ladle shroud has a high velocity. After the liquid steel impacts the impact pad at the bottom of the tundish at a high speed, the reaction force at the bottom of the impact pad flows to the liquid surface of the tundish and continues to flow along the vertical direction of the weir after being blocked by the porous baffle wall. After flowing out of the diversion hole on the porous baffle wall, the liquid steel continues to flow to the far end. Along different deflector holes, the liquid steel with different paths forms two large eddies at the submerged nozzle of the second strand and the third strand in the pouring zone respectively, which increases its residence time in the tundish, which is conducive to

the floating of inclusions, but may also lead to the rapid outflow of part of the liquid steel from the second strand, thus shortening the residence time in the tundish.

### 3.2. Analysis of Tundish RTD Curve and Flow Field in Fewer Strands Casting

In the case of fewer strands casting, the number of submerged nozzle on one side of tundish is reduced, resulting in different number of submerged nozzle on both sides, so it cannot be regarded as symmetrical. According to the RTD curve and simulation analysis, it is found that the result of the other side of the tundish in the case of fewer strands casting is not much different from that in the case of normal casting, so this paper only conducts comparative analysis on the side with fewer strands. The RTD curves of the fewer strands side under different conditions are shown in Figure 5.

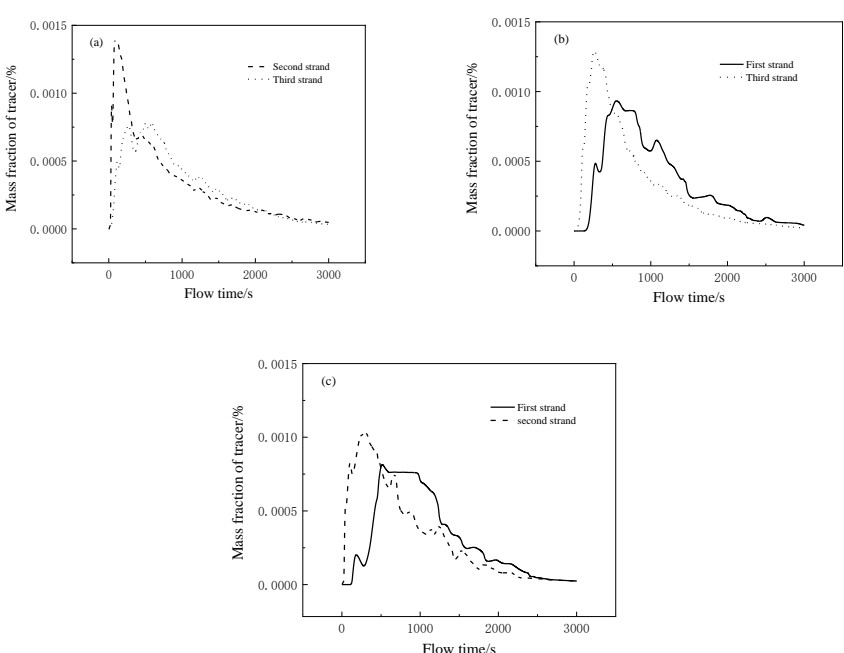

**Figure 5.** RTD curves of tundish for fewer strands casting: scheme 1 (**a**); scheme 2 (**b**); scheme 3 (**c**).

It can be seen from Figure 5a that after the first strand is closed, the RTD curve trend of the other two flows in the tundish is similar to that in normal pouring, but the third strand has a double peak phenomenon, which can be seen that the closing of the first strand worsens the flow field of the tundish. The peak concentration of flow 3 in Figure 5b is significantly higher than that of normal casting in Figures 4a and 5a, indicating that the mixing efficiency of liquid steel in tundish decreases. In contrast, the second strand with a higher peak concentration originally decreased significantly in Figure 5c, and the RTD curves of the two flows coincided with those of the other two schemes with good consistency.

Parameters of tundish flow field characteristics are obtained according to RTD curves in Figures 4 and 5, and the results are shown in Table 4. Compared with normal pouring, the average residence time of each flow in the fewer strands side of each scheme is increased, but the difference between the response time and peak concentration time between the two flows is significantly increased. The response time of the first strand and the third strand in scheme 2 is 161 s and 40 s respectively, the consistency of liquid steel is poor, and the flow characteristics of each flow of liquid steel are not consistent enough, which has a certain impact on the quality of finished products. In terms of dead zone volume fraction, the dead zone volume fraction of the first strand increased from 10.9% to 17.9% and 16.4% respectively for schemes 2 and 3; the dead zone volume fraction of the second strand increased from 8.5% to 14.8% and 9.8% respectively for schemes 1 and 3; and the dead zone volume of the third strand increased from 8.0% to 16.4% and 10.0% respectively

for schemes 1 and 2, it can be seen that the dead zone volume fraction of each flow in each scheme of low-flow casting increases to varying degrees, indicating that fewer strands casting does have a negative impact on the flow field, which is not conducive to the removal of inclusions.

**Table 4.** Numerical result of RTD curves in tundish with each scheme.

| Scheme | Opened Strand | Response Time/s | Peak Concentration Time/s | Average Residence Time/s | Volume Fraction of Dead Zone/% | Volume Fraction of Plug Zone/% | Volume Fraction of Well-Mixed Zone/% |
|---|---|---|---|---|---|---|---|
| Normal casting | 1 | 66 | 253.5 | 806 | 10.9 | 19.8 | 69.3 |
| | 2 | 16 | 68.5 | 648 | 8.5 | 6.5 | 85.0 |
| | 3 | 34 | 162.5 | 705 | 8.0 | 13.9 | 78.1 |
| Scheme 1 | 2 | 20.5 | 81.5 | 802 | 14.8 | 6.4 | 78.8 |
| | 3 | 30.5 | 585.0 | 941 | 16.4 | 32.7 | 50.9 |
| Scheme 2 | 1 | 161 | 553.0 | 1 050 | 17.9 | 34.0 | 48.1 |
| | 3 | 40 | 267.5 | 757 | 10.0 | 20.3 | 69.7 |
| Scheme 3 | 1 | 123.5 | 517.5 | 1 062 | 16.4 | 30.2 | 53.4 |
| | 2 | 22 | 318.5 | 767 | 9.8 | 22.2 | 68.0 |

Figures 6 and 7 show the flow diagram of liquid steel in tundish of three schemes under fewer strands casting and the velocity cloud diagram of outlet section. It can be seen that the movement characteristics of liquid steel in the fewer strands casting zone of the three schemes are obviously different. During normal casting, when the liquid steel enters the pound region along different deflect holes on the porous baffle wall, it first converges above the second strand and forms two large eddies at the submerged nozzle of the second strand and the third strand. For scheme 1, as shown in Figures 6a and 7a, when the liquid steel enters the tundish casting region, the characteristics are basically the same as those in normal casting. However, combined with the RTD curve, it can be seen that although the response time of the liquid steel in the second and third strands is basically the same as that in normal casting, the peak concentration time of the third strand is delayed from 162.5 s to 585.0 s, according to the analysis, due to the closure of the first strand, a part of the liquid steel cannot flow out when it reaches the submerged nozzle of the first strand, but can only return. The distance between the first strand and the third strand is too far. When a part of the returned liquid steel reaches the submerged nozzle of the third strand after a long time, it flows out from the submerged nozzle of the third strand together with the liquid steel just entering the pound region, resulting in the delay of the peak concentration time. For scheme 2, as shown in Figures 6b and 7b, after the liquid steel enters the pound region, due to the closure of the second strand, the liquid steel cannot flow out when it reaches the submerged nozzle of the second strand. It can only continue to advance, stay in the tundish for a longer time, and finally flow out from the submerged nozzle of the first and third strand, which is the reason why the residence time of the fewer strands pound region is generally longer. Combined with the RTD curve, the response time of flow 1 and flow 3 in Scheme 2 is almost the same as that of normal pouring, and the peak concentration time is delayed from 253.5 s and 162.5 s to 553.5 s and 267.5 s, which verifies the conclusion that the residence time of liquid steel on the side with less flow is longer. For scheme 3, as shown in Figures 6c and 7c, the liquid steel continues to move in the pouring zone after entering the pouring zone and reaching the submerged nozzle of the third strand, and flows slowly to the first strand along the bottom of the tundish, resulting in the delayed change of peak concentration. Due to the uneven distribution of submerged nozzle in the three schemes, part of the liquid steel which should enter the mold cannot flow out in time and flows slowly in the tundish, which leads to the consistency of each flow and the increase in dead zone volume. Consistent with the content of the literature [18], the consistency of strands was poor, and the liquid steel cannot be added to many regions on the left side (only two working outlets) of the casting area in time. The velocity in these regions, accompanying serious internal circulation, was very slow, which could cause these regions

to easily become cold steel regions, where low-temperature liquid steel accumulated, and then the low-temperature liquid steel entering into the working outlets could cause nozzle clogging. From the point of view of the response time of each scheme, the response time of the second strand and third strand is not much different from origin scheme, but the first strand is significantly increased, the response time of first strand in scheme 2 and scheme 3 increased from 66 to 161 and 123.5, respectively. This is because the distance from the first strand to the ladle shroud is the furthest, which aggravates the consequences of slow liquid steel flow.

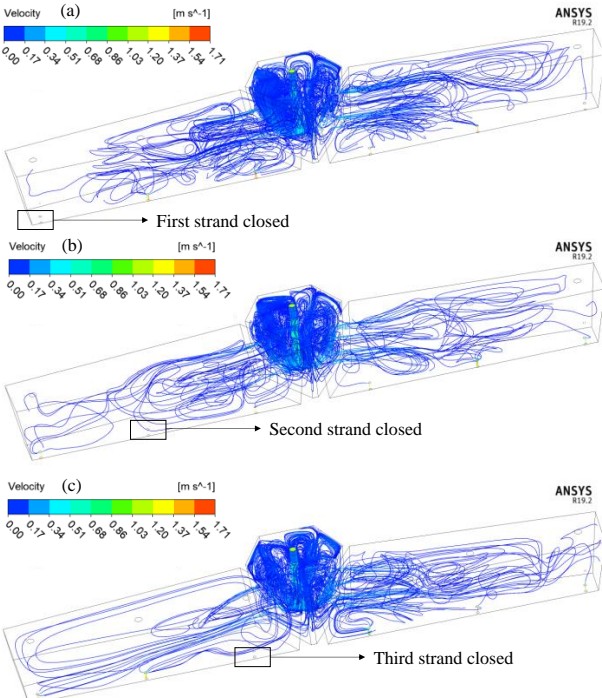

**Figure 6.** Liquid steel flow diagram in tundish for fewer strands casting: scheme 1 (**a**); scheme 2 (**b**); scheme 3 (**c**).

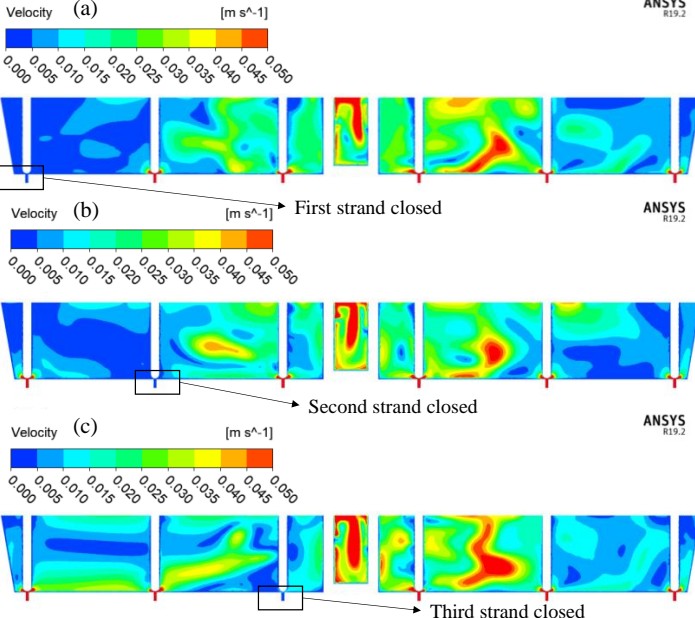

**Figure 7.** Outlet section velocity cloud of liquid steel in tundish for fewer strands casting: scheme 1 (**a**); scheme 2 (**b**); scheme 3 (**c**).

### 3.3. Analysis of Inclusion Removal Rate

### 3.3.1. Inclusion Removal Rate of Total Tundish

Figure 8 shows the inclusion removal efficiency of tundish in each scheme of normal casting and fewer strands casting. It can be seen that the removal rates of inclusions with a particle size of 10 μm are 12.4%, 14.7%, 12.4%, and 14.7% respectively for normal casting and scheme 1, 2, and 3, and the removal rates of inclusions with size of 90 μm are 95.6%, 93.8%, and 93.1% and 95.1% respectively, a small difference. For the inclusion with size of 30 μm, scheme 2 decreased significantly from 39.1% to 28.0%, scheme 1 and scheme 3 were 36.4% and 42.2%, respectively, with little change. For the inclusion with a particle size of 50 μm, the removal rate under normal casting is 74.2%, the removal rate of scheme 1 is 76.4%, and the removal rate of scheme 2 and scheme 3 is the same, both decreasing to 65.8%. For the inclusion with a particle size of 70 μm, the removal rate of each scheme is lower than that of 93.3% under normal pouring, and the decrease in scheme 1 and scheme 2 is 85.3% and 83.6%, respectively, and scheme 3 is 90.7%. It can be seen that fewer strands casting has a negative effect on the inclusion of 30~70 μm particle size, and scheme 2 has the worst performance on the removal of different particle size inclusions.

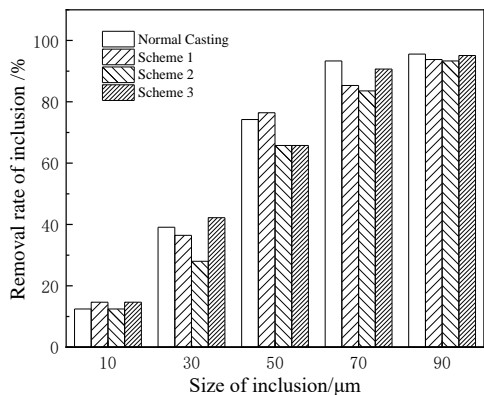

**Figure 8.** Inclusion removal rate of each scheme in tundish.

### 3.3.2. Effect of Fewer Strands Casting on Inclusion Removal

The six-flow bloom tundish studied in this paper is divided into casting region 1 (fewer strands casting region), pound region and casting region 2 by porous baffle wall, as shown in Figure 1c. Since the flow fields of casting region 1 and casting region 2 are different, the influences on inclusion floating removal are also different. Therefore, it is necessary to study the inclusion floating removal rates of the two casting region of each scheme separately.

Figure 9 shows the comparison of floating rate of inclusions in casting region 1, i.e., fewer strands casting region, of each scheme. It can be seen that for large inclusions 70~90 μm, the floating rate of inclusions in the fewer strands casting region of each scheme is significantly lower than that of normal casting, and the removal rates of inclusions in normal casting are 32.0% and 32.0%, respectively, while those in each scheme are about 20.0%. Only scheme 1 with a particle size of 70 μm is 28.4%, which is slightly lower than that of normal casting. For small inclusions with particle size of 10 μm, the inclusion removal rates of normal casting and each scheme are 2.7%, 2.2%, 1.8%, 2.7%, respectively, with little difference. For the inclusion with a particle size of 30 μm, the removal rate of scheme 2 decreased to 5.8% compared with 11.1% of normal casting, which was more obvious, and there was no change in other schemes. For the inclusion with a particle size of 50 μm, the removal rate of scheme 1 is 29.3%, even higher than the normal casting 25.8%, and the other schemes are also lower than the normal casting 19.1% and 17.3%, respectively. It can be concluded that the inclusion removal effect of scheme 1 is more ideal than other schemes in the case of fewer strands casting.

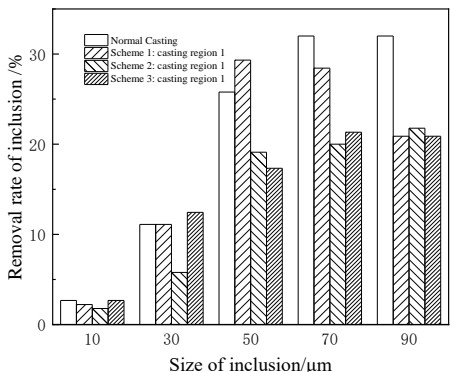

**Figure 9.** Removal rate of inclusion in fewer strands casting area of each scheme.

In order to clarify the influence of fewer strands casting on the removal of inclusions in tundish, the inclusion removal rates of different casting zones in the same scheme should also be compared. Figure 10 shows the comparison of inclusion removal rate between casting region 1 and casting region 2 in each scheme. It can be seen that the inclusion removal rate of casting region 2 in each scheme is basically consistent with that of normal casting, indicating that the fewer strands casting in casting region 1 has almost no influence on the impurity removal efficiency of casting zone 2. As for the inclusion removal rate of casting region 1, when the inclusion particle size is 30–90 μm in scheme 2, the removal rate of casting region 1 has a significant decrease compared with that of casting region 2. When the inclusion particle size is 50–90 μm in scheme 3, the removal rate of casting region 1 has the same decreasing trend compared with that of casting region 2. For scheme 1, the inclusion removal rate of casting region 1 only decreased when the particle size was 90 μm.

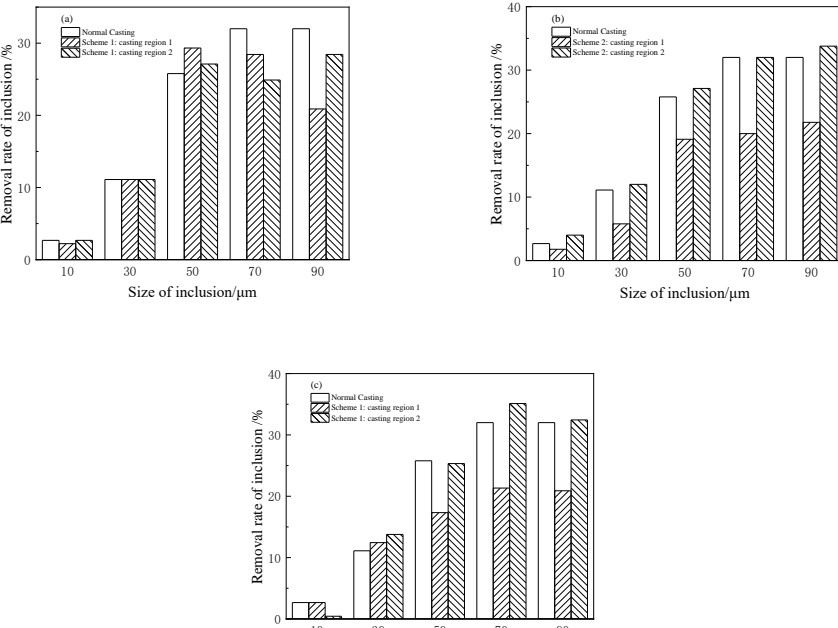

**Figure 10.** Comparison of inclusion removal rate between casting region 1 and 2 in each scheme: scheme 1 (**a**); scheme 2 (**b**); scheme 3 (**c**).

In conclusion, scheme 1 has the least influence on inclusion removal when the tundish is casting with less strands.

### 3.4. Gas Curtain Position under Optimal Strand Closing Scheme

According to the above results, it can be known that the operation of strand closing will increase the volume of the dead zone in the tundish, and the removal of small inclusions is also at a poor level, so it is considered to set a gas curtain in the tundish. The simulation results are shown in Figures 11 and 12 and Table 5. Only the best scheme is shown here. The gas curtain is installed at position 2.

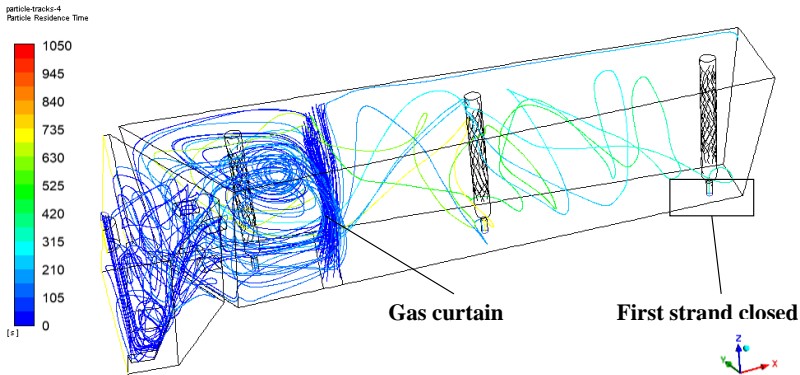

**Figure 11.** The trajectory of the inclusions in the tundish.

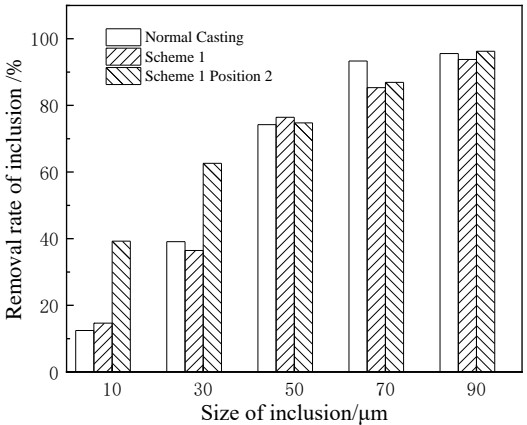

**Figure 12.** Inclusion removal rate in tundish with each scheme.

**Table 5.** Numerical result of RTD curves in tundish with each scheme.

| Scheme | Strand Number | Response Time/s | Peak Concentration Time/s | Average Residence Time/s | Volume Fraction of Dead Zone/% | Volume Fraction of Plug Zone/% | Volume Fraction of Well-Mixed Zone/% |
|---|---|---|---|---|---|---|---|
| Normal casting | 1 | 66 | 253.5 | 806 | 10.9 | 19.8 | 69.3 |
| | 2 | 16 | 68.5 | 648 | 8.5 | 6.5 | 85.0 |
| | 3 | 34 | 162.5 | 705 | 8.0 | 13.9 | 78.1 |
| Scheme 1 (no argon) | 2 | 20.5 | 81.5 | 802 | 14.8 | 6.4 | 78.8 |
| | 3 | 30.5 | 585.0 | 941 | 16.4 | 32.7 | 50.9 |
| Scheme 1 (Position 2) | 2 | 31.3 | 330.3 | 823.1 | 13.9 | 21.9 | 64.0 |
| | 3 | 49.3 | 351.3 | 912.1 | 14.1 | 21.9 | 63.9 |

As can be seen from the trajectory of inclusions in Figure 11, when the inclusions enter the casting region from the deflector hole, the gas curtain will intercept most of the inclusions, making it difficult for them to reach the area with slow flow, and quickly rise before the porous baffle wall and the gas curtain. The gas curtain will also blow the liquid steel to make it rise rapidly, and drive the inclusions to float, so as to achieve the purpose of removing inclusions.

Table 5 shows the comparison between the original tundish and the tundish with each schemes, include first strand closed, and on the basis of closing the first strand, the gas curtain is set at position 2. After setting the gas curtain, the dead zone volume fraction of the second strand and the third strand in the tundish decreases from 14.8% and 16.4% to 13.9% and 14.1%, respectively, with better consistency. As can be seen from the removal rate of inclusions in Figure 12, the gas curtain has a good removal rate on inclusions with sizes of 30 μm and below, and the removal rate of inclusions with sizes of 10 μm and 30 μm increases from 14.7% and 36.4% to 39.2% and 62.6%, respectively. For inclusions 50 μm and above, the gas curtain brings little changes.

## 4. Conclusions

From the performed numerical simulations it can be found that:

(1) Compared with normal casting, the flow field in the tundish deteriorates in the process of fewer strands casting, and the dead zone volume increases somewhat compared with normal casting. The dead zone volume fraction calculated separately by each flow in each scheme increases by about 2~10%.

(2) Under the condition of fewer strands casting, the inclusion removal rate of the tundish with the particle size of 30–70 μm decreased to different degrees, and the inclusion removal effect was the worst when the second strand is closed. The removal rates of inclusions with diameter 30, 50, and 70 μm in the tundish decreased from 39.1%, 74.2%, and 93.3% to 28.0%, 65.8%, and 83.6%, respectively.

(3) When the first strand is closed, the inclusion removal consistency of casting region on both sides of tundish is good; when the second or third strand is closed, the consistency of inclusion removal with particle size of 50~90 μm in the pound region on both sides of tundish is poor.

(4) When the tundish needs to close a strand to adapt to the production rhythm, closing first strand has the least influence on the removal of inclusions in the tundish. The removal rates of inclusions at 10, 30, 50, 70, and 90 μm changed from 12.4%, 39.1%, 74.2%, 93.3%, and 95.6% to 14.7%, 36.4%, 76.4%, 85.3%, and 93.8%, respectively.

(5) Setting an gas curtain in the tundish of the first strand can reduce the dead zone volume. The dead zone volume of the second and third strand is reduced from 14.8% and 16.4% to 13.9% and 14.1%, respectively. The removal of 10 μm and 30 μm inclusions increased from 14.7% and 36.4% to 39.2% and 62.6%.

**Author Contributions:** Conceptualization, D.C. and M.L.; methodology, X.W. and S.W.; software, X.W. and S.W.; validation, X.W. and H.H.; formal analysis, X.W. and S.W.; investigation, X.X. and C.W; resources, X.X. and C.W.; data curation, X.W.; writing—original draft preparation, X.W.; writing—review and editing, X.W.; visualization, X.W.; supervision, D.C. and M.L.; project administration, D.C. and M.L. All authors have read and agreed to the published version of the manuscript.

**Funding:** This work is financially supported by the National Natural Science Foundation of China (NSFC), Project Nos. 52274320 and 52074053.

**Data Availability Statement:** Data sharing is not applicable.

**Acknowledgments:** I would particularly like to acknowledge team members for their support and assistance.

**Conflicts of Interest:** The authors declare no conflict of interest.

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
