# Peer review of "Flow Behavior of Liquid Steel in Fewer Strands Casting of Six-Strand Bloom Tundish"

_metals, doi:10.3390/met13040706_

Round 1

Reviewer 1 Report

The manuscript „Flow behavior of liquid steel in fewer strands casting of six strand bloom tundish“ brings interesting results in the simulation of casting of liquid steel and the effect on the occurrence of unwanted inclusions. It was found that fewer strands casting has a negative effect on the inclusion of 30-70 μm particle size. Furthermore, the influence of closing of various strands on the inclusion removal effect was discussed.

 Major remark:

 It would be appropriate to support the calculations in this article with at least one or two experiments of randomly selected samples that partially meet the selected assumptions. The manuscript is based on computational simulations, which are calculated on the basis of certain assumptions summarized clearly in chapter 2.2. Various simplifications are also assumed - for example, that they are spherical inclusions. Or in chapter 2.4 (please correct the capital letter in the title) it is mentioned in the first sentence that "According to the sampling and testing, the inclusion of silicon killed steel in the pouring process of tundish is mainly silica, and the size range is mainly 10-90μm." So, that it is mainly nonmetallic inclusions in liquid steel. It would be good to have experimental results from at least one sample to confirm these assumptions / results obtained by calculations.

 Minor comments:

 - DPM – specify at the first mention that it is a Discrete Phase Model

 - "computational fluid dynamics (CFD) software ANYSY-Meshing" - software ANSYS is used for CFD

 - a clearer description of the images, for example: Diagram of tundish structure.(a)vertical view; (b) front view; (c) Schematic diagram of fewer strands casting. ---> Diagram of tundish structure: vertical view (a); front view (b). Schematic diagram of fewer strands casting (c).

 - the formula for Fluid resistance ?? – it would be appropriate to explain the derivation, or the origin of the formula; also the description mentions m2 and the formula m

 - gas – explanation of what kind of gas it is. Information about gas (air curtain / argon in the gas curtain) would be appropriate to specify.

- in table 4, for clarity, it would be appropriate to insert "Opened strand" (or similar) instead of "Strand number".

Author Response

Dear reviewer:

I am honored that you reviewed my paper, and in response to the questions you raised, I have made the following modifications:

Q1: It is mainly nonmetallic inclusions in liquid steel. It would be good to have experimental results from at least one sample to confirm these assumptions / results obtained by calculations.

A1:Now I cite the literature that describes the type and size of inclusions in the same tundish.

Q2: DPM – specify at the first mention that it is a Discrete Phase Model.

A2: Now I specified at the first mention that it is a Discrete Phase Model.

Q3: "computational fluid dynamics (CFD) software ANYSY-Meshing" - software ANSYS is used for CFD.

A3: Now it is revised.

Q4: A clearer description of the images, for example: Diagram of tundish structure.(a)vertical view;   (b) front view;   (c) Schematic diagram of fewer strands casting.   --->   Diagram of tundish structure: vertical view (a);   front view (b). Schematic diagram of fewer strands casting (c).

A4: I modified the description of all the images according to the format.

Q5: The formula for Fluid resistance ?? – it would be appropriate to explain the derivation, or the origin of the formula; also the description mentions m2 and the formula m.

A5: I cited relevant literature to explain the formula.

Q6: Gas – explanation of what kind of gas it is.   Information about gas (air curtain / argon in the gas curtain) would be appropriate to specify.

A6: Now I explained the gas is argon, when the gas curtain was first mentioned.

Q7: In table 4, for clarity, it would be appropriate to insert "Opened strand" (or similar) instead of "Strand number".  

A7: Now it is revised.

Reviewer 2 Report

The paper described the flow behavior of liquid steel in Tandish. However, the results presented are precise, but the plot's explanation can be further improved so that the quality of the article can be further enhanced. 

The problem definition can be further explained by stating the research gap that you have observed from your literature survey.

The results obtained from your research work can be compared with the existing published research articles.

The reasons for different response time for different schemes need to be explained clearly.

Justify the removal rate of inclusion by comparing different schemes.

The paper can be accepted with minor corrections

Author Response

Dear reviewer:

I am honored that you reviewed my paper, and in response to the questions you raised, I have made the following modifications:

Q1: The problem definition can be further explained by stating the research gap that you have observed from your literature survey.

A1: I cite more references to illustrate the research.

Q2: The problem definition can be further explained by stating the research gap that you have observed from your literature survey.

A2: I cited the literature to compare with the research content on lines 323-328.

Q3: The reasons for different response time for different schemes need to be explained clearly.

A3:I explain the reasons for different response time for different schemes on lines 329-333.

Q4: Justify the removal rate of inclusion by comparing different schemes.

A4: I think I explained the removal rate of inclusion by comparing different schemes in 3.3.

Reviewer 3 Report

 Dear authors,

Thank you very much for sending us the manuscript. The studies show an interesting effect of the activated strands on the distribution of the impurities. The flow affected by different combinations of activation of the strands gives insight into the impact on the distribution of impurities as a function of particle size. This is impressive for me since this is not my area of expertise. Regarding the manuscript, I have no suggestions for improvement other than some corrections, but I have some comments on the structure or wording I did not clearly understand.

Corrections

136-148: A literature reference would be nice.

336: Should it be a plan or scheme in "..... Plans 1, 2, and 3...."

General comments:

The labeling of the axes in the figures could be larger.

The assumptions in 2.2 and 2.2.2 can perhaps be summarized in one point

I could not find the time values given in sections 233-239 in the figures.

The flow velocities are colored so that in the range of low velocities, the color scale shows only a blue color. Figures 4 and 6, therefore, show almost exclusively blue lines. Velocity differences in the tundish are thus no longer discernible.

Some sentences are relatively long and, therefore, difficult to understand on first reading.

I did not understand the first paragraph in section 3.3. (lines 255-261)

Author Response

Dear reviewer:

I am honored that you reviewed my paper, and in response to the questions you raised, I have made the following modifications:

Q1: Corrections 136-148: A literature reference would be nice.

A1: Now literature reference is cited.

Q2: 336: Should it be a plan or scheme in ".....  Plans 1, 2, and 3...." 

A2: ".....  Plans 1, 2, and 3...." is modified to ".....  Scheme 1, 2, and 3...."

Q3: The labeling of the axes in the figures could be larger.

A3: Now the labeling of the axes in the figures are larger.

Q4: The assumptions in 2.2 and 2.3.2 can perhaps be summarized in one point I could not find the time values given in sections 233-239 in the figures.

A4: The assumptions in 2.2 and 2.3.2 are summarized in one point. The time values given in sections 233-239 in the figures are calculated by formula (8)-(13), and the data is taken from figure 4(a).

Q5: The flow velocities are colored so that in the range of low velocities, the color scale shows only a blue color.  Figures 4 and 6, therefore, show almost exclusively blue lines.  Velocity differences in the tundish are thus no longer discernible.

A5: Figures 4 and 6 mainly display the streamline distribution of liquid steel in tundish, and Figure 7 mainly display velocity differences.

Q6: I did not understand the first paragraph in section 3.3.  (lines 255-261)

A6: What I want to express in this section is: the tundish is distributed symmetrically, with three strands on the left and right sides. When pouring with less strand, only one side has less strand (2 strands), and the other side is still normal (3 strands). So only the less-strand side is studied.

Round 2

Reviewer 1 Report

Dear Author,

I recommend article for publication.